# Convolutional neural networks improve species distribution modelling by capturing the spatial structure of the environment

Benjamin Deneu[1,2,3]*, Maximilien Servajean[2,4], Pierre Bonnet[3,5], Christophe Botella[1,2,3], François Munoz[6], Alexis Joly[1,2]

**1** Inria, Montpellier, France, **2** LIRMM, Univ Montpellier, CNRS, Montpellier, France, **3** AMAP, Univ Montpellier, CIRAD, CNRS, INRAE, IRD, Montpellier, France, **4** AMIS, Université Paul Valéry Montpellier, CNRS, Montpellier, France, **5** CIRAD, UMR AMAP, Montpellier, France, **6** LIPHY, Université Grenoble Alpes, Grenoble, France

* benjamin.deneu@inria.fr

**Data Availability Statement:** All relevant data are available from the cnn-sdm gitlab project (url: https://gitlab.inria.fr/bdeneu/cnn-sdm) along with examples codes.

## Abstract

Convolutional Neural Networks (CNNs) are statistical models suited for learning complex visual patterns. In the context of Species Distribution Models (SDM) and in line with predictions of landscape ecology and island biogeography, CNN could grasp how local landscape structure affects prediction of species occurrence in SDMs. The prediction can thus reflect the signatures of entangled ecological processes. Although previous machine-learning based SDMs can learn complex influences of environmental predictors, they cannot acknowledge the influence of environmental structure in local landscapes (hence denoted "punctual models"). In this study, we applied CNNs to a large dataset of plant occurrences in France (GBIF), on a large taxonomical scale, to predict ranked relative probability of species (by joint learning) to any geographical position. We examined the way local environmental landscapes improve prediction by performing alternative CNN models deprived of information on landscape heterogeneity and structure ("ablation experiments"). We found that the landscape structure around location crucially contributed to improve predictive performance of CNN-SDMs. CNN models can classify the predicted distributions of many species, as other joint modelling approaches, but they further prove efficient in identifying the influence of local environmental landscapes. CNN can then represent signatures of spatially structured environmental drivers. The prediction gain is noticeable for rare species, which open promising perspectives for biodiversity monitoring and conservation strategies. Therefore, the approach is of both theoretical and practical interest. We discuss the way to test hypotheses on the patterns learnt by CNN, which should be essential for further interpretation of the ecological processes at play.

## Author summary

Species distribution models aim at linking species spatial distribution to the environment. They can highlight the ecological preferences of species and thus predict which species are

**Funding:** This study was possible thanks to the financial support of the Labex Numev and the French National Research Agency under the Investments for the Future Program, referred as ANR-16-CONV-0004. The funders had no role in study design, data collection and analysis, decision to publish, or preparation of the manuscript.

**Competing interests:** The authors have declared that no competing interests exist.

likely to be present in a given environment. These models are used in many scenarios such as conservation plans or monitoring of invasive species. The choice of model and the environmental data used have a strong impact on the model's ability to capture important information. Specifically, state-of-the-art models generally use a punctual environment and do not take into account the environmental context or neighbourhood. Here we present a species distribution model based on a convolutional neural network that allows the use of large scale data such as spatialized environmental data including the environmental neighbourhood in addition to the punctual environment. We highlight the interests and limitations of this method as well as the importance of the environmental context in learning about species distributions.

## Introduction

Species Distribution Models (SDM) characterize the relationship between the environment and species occurrences, depending on their ecological niches [1]. The ecological niche is multidimensional, and involves factors playing in a complex fashion (non linear) and at multiple spatial scales. Therefore, capturing the complexity of ecological niches remains a major challenge when designing SDMs. For practical reasons (data required), most Species Distribution Models (SDMs) are correlative methods relating known species occurrence data to potential environmental predictors [2–7]. Popular examples of such correlative methods include MAXENT (used for instance in [8–10]), random forest (used for instance in [11]) and boosted regression trees (used for instance in [12–14]).

Some recent SDMs use deep neural networks to better address the complexity of ecological niches. Devising SDMs based on neural networks is not new [15, 16], but earlier models usually integrated a single hidden layer network. However, deep neural networks architectures are suited to efficiently approximate hierarchical functions which compose local constituents functions, *i.e.* with low dimensional input [17]. For such response functions, it exists a theoretical guarantee that deep neural network architectures outperform one layered architectures, *i.e.* they yield higher statistical generalization power. Recent advances in deep learning have allowed training much deeper neural networks and acknowledging more complexity in the way environment shapes ecological niches [18]. Key advantages of deep learning are that (i) it allows characterizing complex structuring of ecological niche depending on multiple environmental factors, (ii) it can learn niche features common to a large number of species, and thus grasp the signatures of common ecological processes and improve SDM predictions across species [18, 19].

A specific class of neural networks initially proposed in [20], named Convolutional Neural Networks (CNN), has very recently been proposed for SDM [19, 21]. A specific property of CNN is that they rely on spatial environmental tensors rather than on local values of environmental factors. These tensors represent the spatial realisation of environmental factors around each point. Unlike other SDM approaches, CNN-based SDMs (CNN-SDMs) can use this very large input data and therefore potentially capture richer information than in punctual vectors. CNNs were originally designed for image classification [20] and proved to outperform any other statistical or machine learning methods in the task of learning complex visual patterns. Indeed there architecture is based on small visual filters that can learn to recognise structural patterns in high dimensional input data such as images. CNN-SDMs should thus be suited to represent how complex ecological niches and spatial dynamics determine the distribution of many species in a region. [19, 21] have shown that CNN-SDMs can improve predictive

performance in SDMs compared to punctual models (models using punctual vectors of environmental data instead of tensors).

We hypothesized that a higher predictive ability of CNN-SDMs should result from the fact that they can grasp the influence of habitat availability, spatial heterogeneity and spatial structure the environment around location, apart from the environmental conditions at the precise location. To test this hypothesis, we designed a set of experiments in which the neural network used or not specific features in the local landscape, such as the spatial structure of environmental variables, or their variability. Moreover, we qualitatively analysed the neuron activations (*i.e.* the numerical realisation obtained for a given input) of the last layer of the network, from which the final linear predictor is built. This is done by mapping each neuron in the geographical space.

Based on the results of the experiments, the main insights we provide here are:

1. the main strength of convolutional neural networks (CNN-SDM) is to provide more reliable predictions for the vast majority of species having only few occurrences in the training set.

2. CNN-SDMs allow capturing the spatial structure of the local environment that is richer than just the local statistical distribution of environmental values and that the use of this information explains in large part the predictive superiority of these methods.

## Materials and methods

### Dataset

We analyzed a large presence-only dataset built from the Global Biodiversity Informatics Facility (GBIF, GBIF.org (31 October 2018) GBIF Occurrence Download https://doi.org/10.15468/dl.l4ofpm). It includes 97, 683 occurrences over the French territory, for 4520 plant species which names follow the TAXREF taxonomic reference [22] from the French National Inventory of Natural Heritage [23]. Uncertainty in spatial locations varies from few meters up to 10km. We randomly split the species occurrences into training (90%) and test (10%) sets. In addition, 10% occurrences from the training provided a validation set. For more details on the dataset construction protocole refer to the supporting information S1 Protocole. Fig 1 shows the distribution of the number of occurrences by species, with very few frequent species and many rare species, yielding a long-tail distribution.

We use 33 environmental raster variables over the French territory (Table 1), including climatic, soil, elevation and land cover variables as predictors in species distribution modelling. We split the categorical land cover variable into 45 different layers each describing the presence (1) or the absence (0) of a land-cover class at a given pixel. We then obtained 77 input dimensions instead of 33. Furthermore, rasters contains sea pixels and other undefined values that should be attributed a numerical value. To avoid as much as possible potential errors related to this constraint, we chose a value sufficiently distinct from the other values, here we choose a value under the minimum of the values of valid pixels.

For performing baseline punctual models, we extracted a punctual environmental vector for each occurrences. For performing CNN, we defined spatialized environmental tensors as following. For a given position and a given environmental variable, we first define a matrix including the variable values in $64 \times 64$ pixels around the position. The matrices of the environmental variables are then aggregated into a single 3-dimensional tensor. Thus, for each species occurrence, we obtain an environmental tensor of size $64 \times 64 \times 77$. Note that the geographical extent of each layer of the tensor depends on the resolution of the raster (for

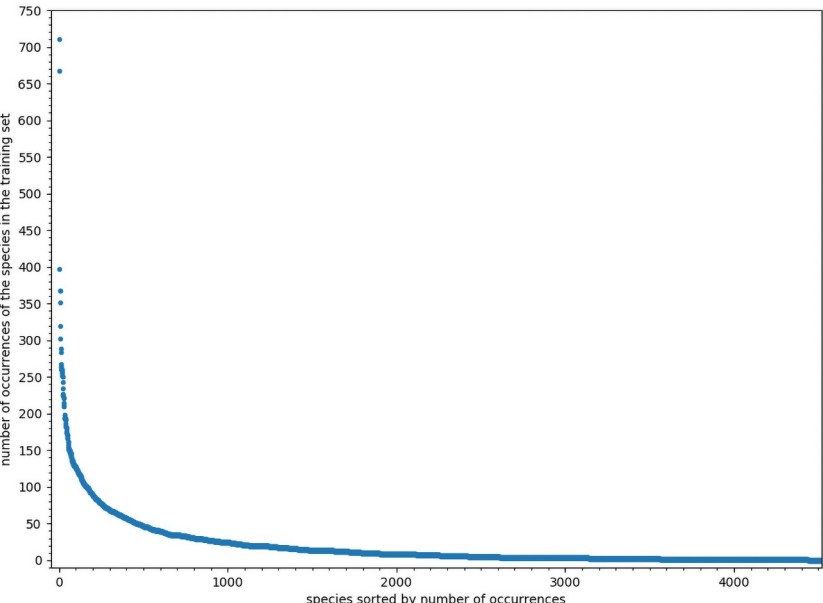

**Fig 1. Occurrences' distribution.** Distribution of occurrences in the training set (including validation set), species are ordered by frequency.

instance, the pixels' resolution corresponds to a 1 km square for the bioclimatic variables, but to a 30 m square for the water proximity). Fig 2 represents an example of such tensor.

In order to be able to compare our predictions for some species with a larger test set we also use INPN occurrence maps for the respective species. These maps are publicly available for each species on the INPN website [23].

## Species distribution models

Since the data set consists of presence-only observations (with no absence data), the model predictions cannot be probabilities of species presence. Some methods in the literature propose the use of pseudo-absences or background models, but it is known that these methods can lead to significant biases [24, 25]. In order to avoid such biases, we propose to work on another type of model than classical Stacked-SDMs [26], Joint-SDMs [27] or multi-species SDMs [28] found in the literature. Rather than predicting the probability of the presence of a species at a given location, the models studied aim at predicting the probability of the species conditionally to the fact that a plant has been observed at a given location. Thus, the output of the models takes the form of a categorical distribution over all possible plant species and the sum over all the species is equal to 1. More formally, given a set of observations $\mathcal{D}$ and an observation $i \in \mathcal{D}$, the prediction $\hat{y}(i)$ is a vector of size $m$ ($m$ = 4520, the number of species), each component $\hat{y}_s(i)$ being the probability that the species of $i$ is $s$. The probabilities $\hat{y}_s(i)$ can then be sorted to obtain a ranked list of the species most likely to occur at a given location. Such type of predictions is useful for a variety of purposes (identification, conservation, prospecting planning, etc.) and is less sensitive to observation bias (in particular, it does not require absence or pseudo-absence data). In the following sub-sections, we describe in detail the different models studied using this paradigm, starting with the CNN (allowing to model the spatial structure) and then moving on to the baseline punctual models.

**Deep convolutional neural network (CNN-SDM).** The main objective of a deep convolutional neural network is twofold. Given some input data $x$ (in our case a $64 \times 64 \times 77$

**Table 1. Environmental variables and description.**

| Name | Description | Nature | Value | Resolution |
|---|---|---|---|---|
| CHBIO_1 | Annual Mean Temp. (mean of monthly) | quanti. | [-10.7,18.4] | 1km |
| CHBIO_2 | Max-temp—min-temp | quanti. | [7.8, 21.0] | 1km |
| CHBIO_3 | Isothermality (100*CHBIO_2/CHBIO_7) | quanti. | [41.1,60.0] | 1km |
| CHBIO_4 | Temp. seasonality (std.dev*100) | quanti. | [302.7, 777.8] | 1km |
| CHBIO_5 | Max Temp of warmest month | quanti. | [6.1,36.6] | 1km |
| CHBIO_6 | Min Temp of coldest month | quanti. | [-28.3,5.4] | 1km |
| CHBIO_7 | Temp. annual range | quanti. | [16.7,42.0] | 1km |
| CHBIO_8 | Mean temp. of wettest quarter | quanti. | [-14.2,23.0] | 1km |
| CHBIO_9 | Mean temp. of driest quarter | quanti. | [-17.7,26.5] | 1km |
| CHBIO_10 | Mean temp. of warmest quarter | quanti. | [-2.8, 26.5] | 1km |
| CHBIO_11 | Mean temp. of coldest quarter | quanti. | [-17.7, 11.8] | 1km |
| CHBIO_12 | Annual precipitations | quanti. | [318.3,2543.3] | 1km |
| CHBIO_13 | Precipitations of wettest month | quanti. | [43.0,285.5] | 1km |
| CHBIO_14 | Precipitations of driest month | quanti. | [3.0,135.6] | 1km |
| CHBIO_15 | Precipitations seasonality (coef. of var.) | quanti. | [8.2,57.8] | 1km |
| CHBIO_16 | Precipitations of wettest quarter | quanti. | [121.6,855.6] | 1km |
| CHBIO_17 | Precipitations of driest quarter | quanti. | [19.8,421.3] | 1km |
| CHBIO_18 | Precipitations of warmest quarter | quanti. | [198,851.7] | 1km |
| CHBIO_19 | Precipitations of coldest quarter | quanti. | [60.5,520.4] | 1km |
| etp | Potential evapo transpiration | quanti. | [133, 1176] | 1km |
| alti | Elevation | quanti. | [-188,4672] | 100m |
| awc_top | Topsoil available water capacity | ordinal | {0,120,165,210} | 1km |
| bs_top | Base saturation of the topsoil | ordinal | {35,62,85} | 1km |
| cec_top | Topsoil cation exchange capacity | ordinal | {7,22,50} | 1km |
| crusting | Soil crusting class | ordinal | [0, 5] | 1km |
| dgh | Depth to a gleyed horizon | ordinal | {20,60,140} | 1km |
| dimp | Depth to an impermeable layer | ordinal | {60,100} | 1km |
| erodi | Soil erodibility class | ordinal | [0, 5] | 1km |
| oc_top | Topsoil organic carbon content | ordinal | {1,2,4,8} | 1km |
| pd_top | Topsoil packing density | ordinal | {1,2} | 1km |
| text | Dominant surface textural class | ordinal | [0, 5] | 1km |
| proxi_eau_fast | <50 meters to fresh water | boolean | {0,1} | 30m |
| clc | Ground occupation | categorial | [1, 48] | 100m |

environmental tensor), it first applies non-linear transformations of the data $z = \phi(x)$, to get a new vectorial representation (henceforth called "feature vectors") of lower dimensionality. Second, it fits a generalized linear model to predict the target value $\hat{y} = f(z)$ (here the conditional probability of species given an observed plant specimen) as a function of the new representation. Since the model is optimized for all species jointly, the learnt representation space $z = \phi(x)$ is common to a large number of species, which stabilizes predictions from one species to another and improves them globally. A deep convolutional neural network is thereby a composition of these two functions:

$$\hat{y} = (f \circ \phi)(x) \tag{1}$$

Both functions are fitted jointly during the training process.

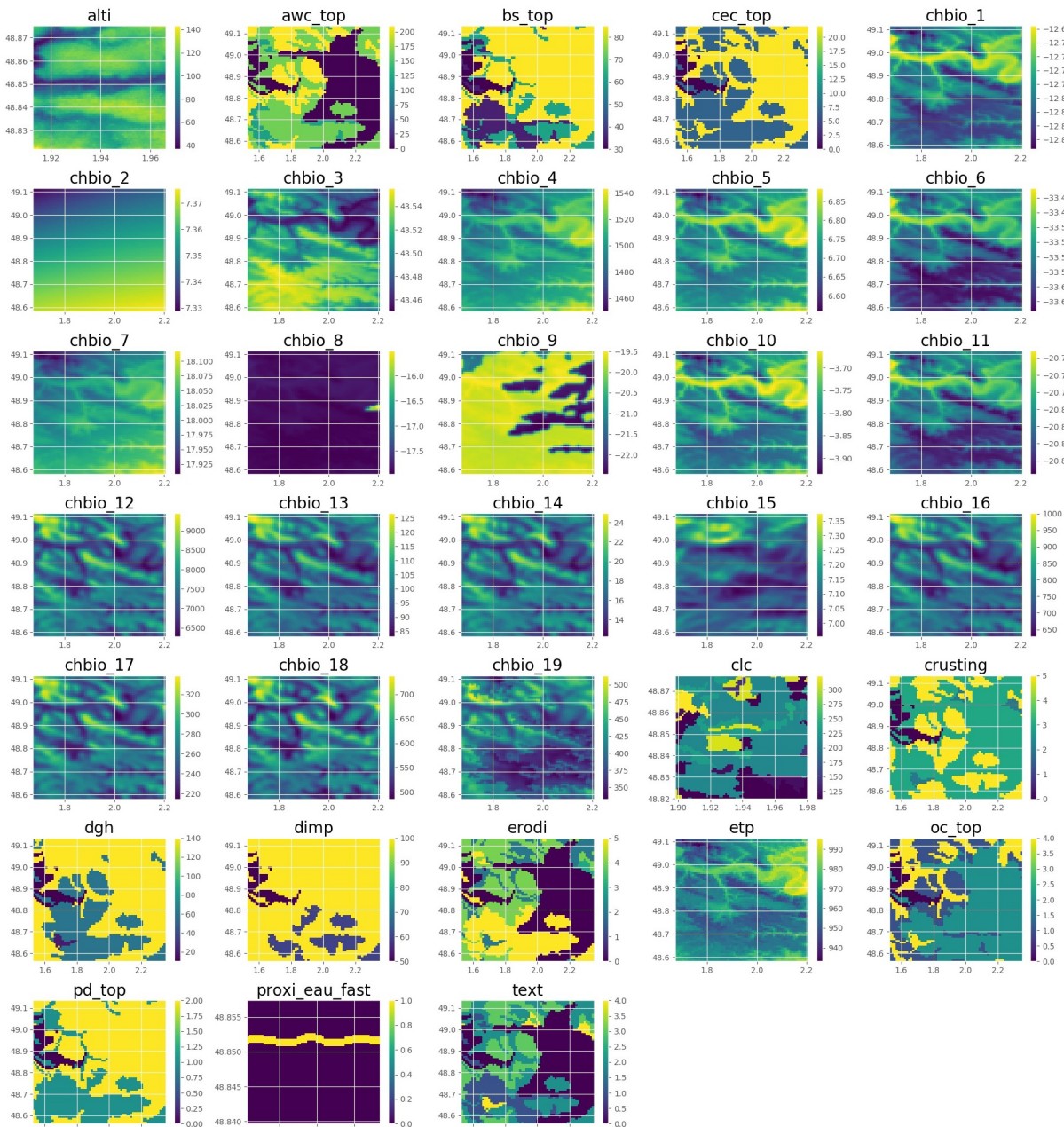

**Fig 2. Tensor example.** Example of tensor extracted from the 33 variables (latitude: 48.848530, longitude: 1.939530). Artificial colors from purple (lowest value) to yellow (highest value).

Categorization tasks based on neural networks are usually performed with a multinomial logistic regression, *i.e.* by defining $f(\cdot)$ as a linear model followed by a softmax link function:

$$\hat{y}_s = f_s(z) = \frac{e^{\beta_s.z+\alpha_s}}{\sum_j e^{\beta_j.z+\alpha_j}} \tag{2}$$

where $\beta_s$ and $\alpha_s$ are the parameters of the linear model learned for species $s$ and the $f_s$s represent the categorical probabilities over the set of species (conidtionally to an observation).

For $\phi(\cdot)$, we chose the Inception v3 architecture [29], with a dropout regularization [30]. This architecture is usually used for image classification, and should be suited to grasp the multi-scale and hierarchical nature of biodiversity patterns. We tested this architecture against alternative options on the validation set, and found it to be the best. The Inception v3 architecture ends up with a feature vector $z$ of size 2048 (to be compared with the 315, 392 dimensions of the input data). Since the feature vectors $z$ are used as input co-variables of the species prediction $f_s(z)$, they are likely to represent synthetic spatial patterns reflecting macroecological and biogeographical structuring common to the selected species. Therefore, the $z$ can also be used further for ecological modelling or prediction tasks.

To fit the model, we used a cross-entropy loss function classically designed for multinomial logistic regression:

$$\mathcal{L}(\hat{y}, y) = -\sum_{k=1}^{K} y_i \, \log \, \hat{y}_i \tag{3}$$

where $\hat{y}_s = f_s(z)$ is the categorical probability of species $s$ given by the model, and $y_s$ is 1 if species $s$ is the correct one and 0 otherwise.

We used a Stochastic Gradient Descent (SGD) to get estimated parameter values, with a decreasing learning rate policy starting at $10^{-1}$, and divided by 10 before epochs 90, 130, 150 and 170. Dropout was set at 0.7 and momentum at 0.9. We processed validation every 5 epochs, and the final model was the one with the highest validation score.

**Punctual deep neural network (DNN-SDM).** We compared the CNN-SDM approach to an alternative deep but non-convolutional neural network model that do not acknowledge spatial patterns around species locations. This model, called DNN-SDM, is a punctual model using environmental vectors (instead of the environmental tensors for the CNN). For sake of comparison, we kept the same architecture as CNN-SDM model, so that differences in performance should be due to the nature of predictors and the way models extracted relevant information from them. Since the architecture requires $64 \times 64$ tensors as input, we designed tensors in which for each layer the central value (the value at the point of the occurrence) is repeated on all pixels of the layer. We used the same training procedure as for CNN-SDM, but with a dropout set to 0.5.

**Boosted trees (BT).** We compared the above models to a state-of-the-art boosted tree model. Gradient tree boosting or boosted trees (BT) are a category of gradient boosting algorithms based on decision trees. Simple decision trees are used in ecology [31], because they have two main advantages: (1) they allow to easily take into account data of different types (integers, booleans, reals, categorical, etc.), (2) they are relatively easy to interpret in most cases. BTs correct the low predictive power of single decision trees by the boosting method. BTs have been commonly used in recent years in ecology for their performance [14, 32–34]. Several studies have shown that they provide significant performance gain compared to previous state-of-the-art models [13, 35]. Here we used the gradient tree boosting algorithm implemented within the xgboost python package [36]. We parameterized our model with a tree depth of 2.

**Random forest (RF).** We also compared CNN-SDM models to the popular Random Forest (RF) approach. RFs have proved efficient in many SDM studies [11, 32, 37–41]. Like boosted trees, RFs are based on decision trees. Their interpretation is less straightforward than a single decision tree, but it is possible to assess the importance of each input variable relative to the others. Moreover, RFs are known to have good predictive performances. Here we used the random forest algorithm implemented within the scikit-learn framework [42]. For hyperparameterization, we used 100 trees with a maximum depth of 16.

## Evaluation

As the models studied in this paper are not classical SDMs, the usual metrics (AUC, TSS, etc.) are not necessarily the most suitable. The output of the model is actually a categorical probability distribution, *i.e.* the probability of the species conditionally to an observation. A classic metric for such predictions is the accuracy, i.e. the percentage of observations for which the correct species is predicted. However, this metric is not adapted to the fact that several species can be observed at the same location. It is therefore preferable to use a set-valued version of the accuracy such as the *top-k accuracy* index, *i.e.* the probability that the true species of the observation belongs to the set of $k$ species predicted as most likely by the model. This *top-k accuracy* is averaged per species (using only the occurrences of each species), and then across species. More formally, for an observation $i \in \mathcal{D}$, we defined as $r_i$ the rank of the true species of $i$ in the sorted list of the estimated probabilities $\hat{y}_s(i)$. And for $k \geq 1$, we defined the *top-k accuracy* as:

$$A_k = \frac{\sum_i^n A_k(i)}{n} \tag{4}$$

with $n$ is the number of occurrences in the test set and

$$A_k(i) = \begin{cases} 1 & \text{if } r_i \leq k \\ 0 & \text{else} \end{cases} \tag{5}$$

To avoid giving too much weight to the most frequent species, it is preferable to evaluate the models in terms of scores per species and not per occurrence. Therefore, we defined the *species-wise top-k accuracy* for a particular species $s$ as:

$$SA_{k,s} = \frac{\sum_i^j A_k(i)}{j} \tag{6}$$

With $j$ the number of occurrences of species $s$ in the test set. Then we defined the *mean top-k accuracy* per species by:

$$MSA_k = \frac{\sum_s^n SA_{k,s}(m)}{n} \tag{7}$$

with $n$ the number of species in the test set.

In addition to this primary evaluation metric, we evaluated two other more traditional metrics, the area under curve (AUC) and the true skill statistics (TSS). This additional assessment allows us to compare our metric with those more commonly used in the literature on SDM. In order to use these metrics, we must first define a choice for the generation of the pseudo-absence. Since our data set is highly unbalanced, a uniform random sampling of the occurrences of the other species would result in over-representation of the most frequent species. As these species are often those with the widest or least specific distributions, this choice would penalize the evaluation of other species that may share certain environments where they are present. To avoid this, we proceed to a weighted sampling over the occurrences, each occurrence being weighted by the inverse of the total number of occurrences of the same species in the test. The sum of the weights of all occurrences of a species is then 1. This is equivalent to first select a species at random and then randomly sample one occurrence of this species. For each species, the sampling of pseudo-absence occurrences is done without replacement (*i.e.* one occurrence cannot be used as a pseudo-absence two times for the same species). To

balance the number of presences and pseudo-absences, we sample a pseudo-absence for each presence. Each species then has the same number of presence and pseudo-absence.

Evaluation via the AUC or TSS does not require species prediction to be categorical or rank-based (unlike our primary metric). It is therefore not necessary to compute them on the main output $\hat{y}$ of the CNN model. We can rather use the values of the logits $\beta_s.z + \alpha_s$ computed by the model before the softmax operator (see Eq 2). The global dynamics of these values is not directly reliable for a prediction of probability of presence or density. However, they are more likely to represent the habitat suitability of each species separately than the categorical probabilities $\hat{y}_s$. For the RF model, it is not possible to use the same methodology since such logits do not exist. Thus, by default, we used directly the categorical probabilities $\hat{y}_s$ to compute the AUC and TSS metrics for that model.

To compute the mean AUC per species (MS_AUC), we first compute the AUC score of each species $s$ based on either the logit value $\beta_s.z + \alpha_s$ (for the CNN) or $\hat{y}_s$ (for RF), then we average the AUC values over all species.

Concerning the mean TSS per species (MS_TSS), a similar but slightly different procedure is used. We first scaled each species predictions values to be between 0 and 1. This is done with a min-max scaler on each species over all test set occurrences. This is necessary for the use of logits as predictions value for the CNN because their range is not known and we need to vary a threshold between the min and the max. For the RF, even if the value returned are already probabilities between 0 and 1, they are relative probability with totally different dynamics than a probability of presence. To evaluate each species with the TSS score we need to limit this effect. The min-max scaler transform the value between 0 and 1 that is not directly reliable to a prediction of presence but that allows to have a comparable range of values across species. Finally, to obtain the MS_TSS score we vary the decision threshold (between 0 and 1) for which we consider that the model predicts the presence of the species and we keep the threshold that gives the highest score for each model.

## Ablation study

Unlike punctual environmental vectors, environmental tensors contain information about the environment within the spatial neighborhood. To better identify how this information is used by CNN-SDM, we have developed ablation experiments. During these experiments we degrade the information contained in the input tensors and we learn the model on these degraded tensors. The impact of the degradation on the performance is directly measured on the final score of the trained model. Ablation is a straightforward way to investigate causality in complex systems such as deep neural networks, by evaluating the contribution of specific characteristics to the overall explanation. We used the approach to examine the value of spatial structuring information learnt by CNN-SDMs, in addition of the environmental information captured by classical SDMs. For each model learnt with an ablation transformation, the transformation is applied at each tensor (*i.e.* for each occurrence) during learning and testing. Fig 3 provides an illustration of the various transformations presented below.

**Random rotations.** The first transformation is a random rotation of the input tensor $x$ in one of the four possible spatial directions (*i.e.* 0˚, 90˚, 180˚ and 270˚). It is applied on all layers with the same rotation and on each occurrence, so that the network cannot acknowledge the actual orientation.

**Random permutations.** For this ablation, the transformation of the tensors carried out is a random permutation of the pixels of each layer of the tensors. Again, the transformation is applied to each layer of each tensor (*ie.* for each occurrence). The spatial structure of the

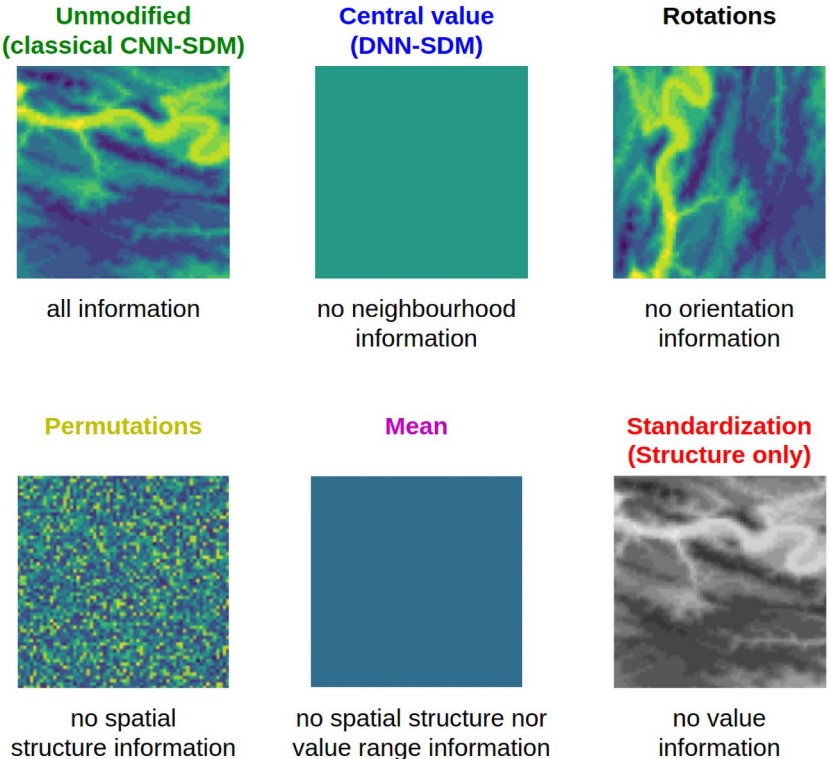

**Fig 3. Ablation study transformations.** Illustration of the ablation study transformations on one layer of a tensor. Artificial colors from purple (lowest value) to yellow (highest value). The standardization transformation is in shades of gray to illustrate the deprivation of true environmental values. The colors of the names correspond to the colors of the respective curves in the results.

environment contained in the tensors is destroyed but the variability of local environmental values is preserved.

**Mean value.** This ablation consists in averaging each tensor layer. As for DNN-SDM, the layers keep the same dimension (64 × 64) but all pixels are identical equal to the average value before transformation. The CNN is then deprived of the information about the spatial structure and variability of the environment initially contained in the tensors. This model uses exactly the same information than a classical punctual SDM model (such as DNN-SDM, BT and RF), but with a spatial regularisation by averaging over the neighborhood. It can thus be compared to the results of DNN, where only punctual environmental values are used, to assess the extent to which the model is sensitive to local environmental value against coarser-grained environmental information.

**Structure only.** The above ablation tests degraded the information about the spatial structure contained in the tensors (spatial structure of the environment around an occurrence). Here we have designed an inverse experiment that retains the information about the spatial structure of the tensors but does not take into account the numerical realization of the environmental values. To achieve this degradation, we apply a standardization to all layer of each environmental tensors. This therefore consists in removing the mean value of the layer and dividing by its standard deviation. CNN then no longer has information about the mean and variance of the local environment around the occurrence. For example, for the altitude, this transformation is equivalent to keeping the spatial shape of the relief (as seen from above) while removing the link with the real altitude and its amplitude.

## Qualitative analysis of the features learnt in CNN-SDMs

We examined the information encoded in the features (*i.e* neurons) learnt by the neural network. More precisely, we analyzed the last layer of the network ($z = \phi(x)$), *i.e* the one on which the final linear predictor $f(\cdot)$ was learnt. The neuron activation values $z_j$ could be interpreted as meta-descriptors of the environment, i.e., as latent variables encoding environmental information used by the final linear predictor. To make the interpretation of these features easier, we mapped them in geographical space. We mapped the features across a spatial grid of 1km-by-1km quadrats over French territory, with neuron activation values $z^q = \phi(x^q)$ calculated at the center of each quadrat $q$. The $z^q$ values were then averaged over larger 10km-by-10km squares, yielding 5400 pixels $s$ with average activation $z_i^s$ for a particular neuron $i$.

We examined the spatial structure of these maps for further interpretation of their environmental and/or geographical nature. For instance, if a neuron was only activated at a specific location, the model likely learnt specific conditions at that location from the set of environmental tensors. If the neuron was activated in large or disjointed areas, it could represent a broader-scale patterns beyond specific local conditions. Such patterns could then represent biogeographic or macro-ecological regions driving species distributions at broad scale.

## Results

### Better predictive performance of CNN-SDMs

Fig 4 compares the predictive performance of the four models, in terms of *mean top-k accuracy* per species for $k \in [1, 100]$ (see section). The CNN-SDM model performed better than alternative punctual models, including DNN-SDM (for all mean comparison for all $k$ between the CNN and another model the p-value is under 0.001). Since DNN-SDM and CNN-SDM models had the same architecture, the performance gain was due to additional information conveyed by spatial environmental tensors, compared to using punctual vectors.

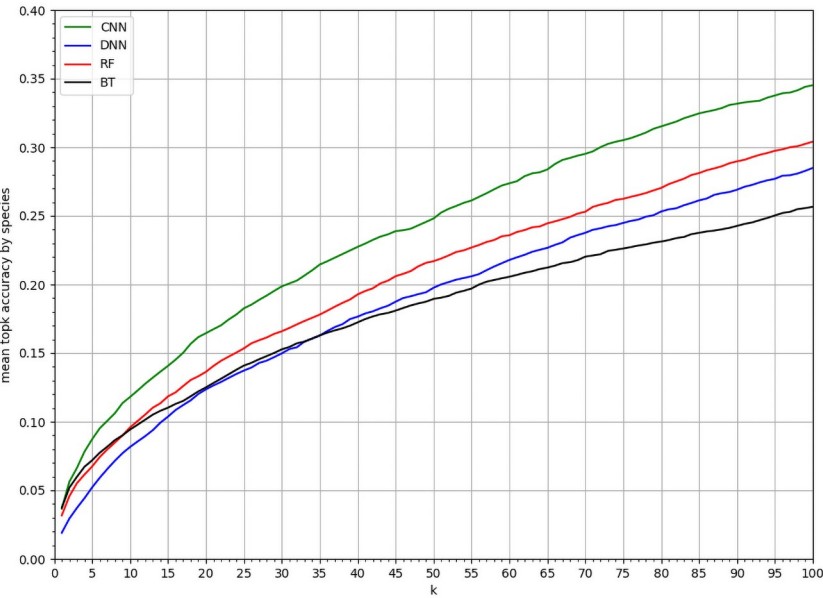

**Fig 4. Performances comparison.** $MSA_k$ of CNN-SDM, DNN-SDM, RF and BT for varying $k$ values within [1, 100].

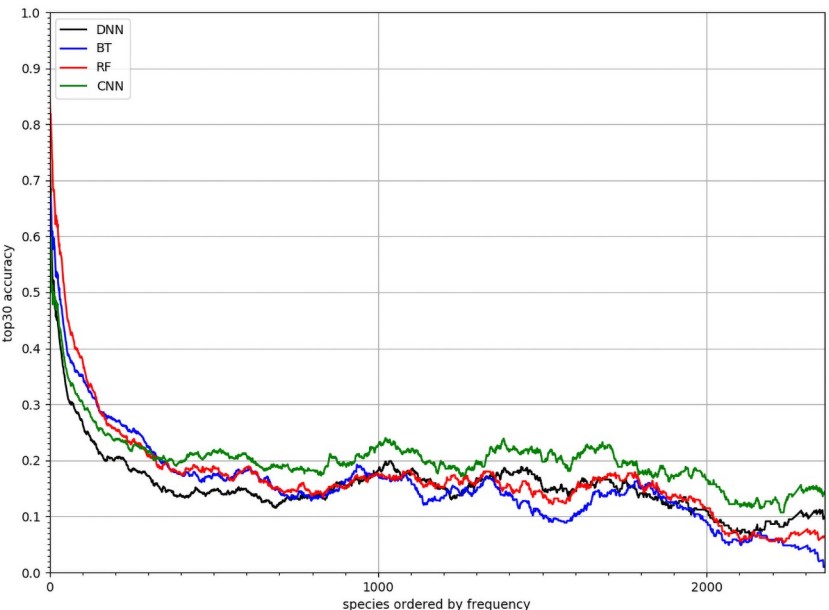

**Fig 5. Performances on rare species $SA_{30}$ of CNN-SDM, DNN-SDM, RF and BT for each species.** Species are sorted by decreasing number of occurrences in the training set. An adaptive moving average was applied to smooth the curves.

## Better performance of CNN-SDMs for rare species

Fig 5 shows the performance achieved by the four models per species ordered by decreasing number of occurrences in the training set (using $SA_{30,s}$). RF and BT models performed better for the most frequent species (which represent the majority of occurrences). However, the CNN-SDM model proved better for less frequent species, which are much more numerous but represent a small proportion of occurrences. As an illustration of this generalization capacity, Figs 6 and 7 displays the predicted responses functions of two species having only one occurrence in the training set. This was done by plotting the predicted linear response $\beta_i.z + \alpha_i$ of each of the two species on a 1 km grid (averaging on a 10km grid for display). For comparison purpose, we also provide for each species a snapshot of the occurrences map publicly available on the web platform of the French National Inventory of Natural Heritage (INPN), managed by the French National Natural History Museum (MNHN). For the two species, the distribution predicted by the CNN-SDM model fitted very well the INPN occurrences (although only

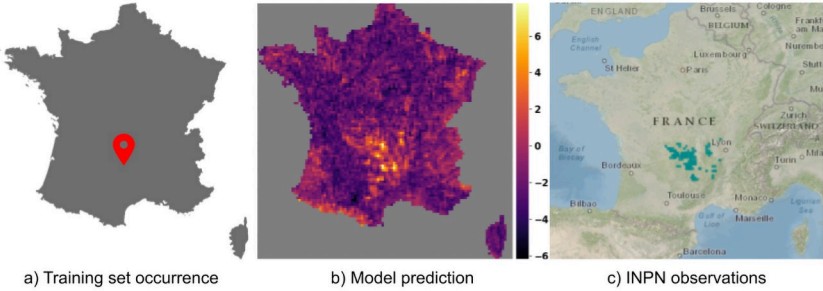

a) Training set occurrence b) Model prediction c) INPN observations

**Fig 6. Prediction of *Senecio cacaliaster* Lam., 1779.** Compare the model prediction to the French National Inventory of Natural Heritage (INPN) occurrences map for *Senecio cacaliaster* Lam., 1779.

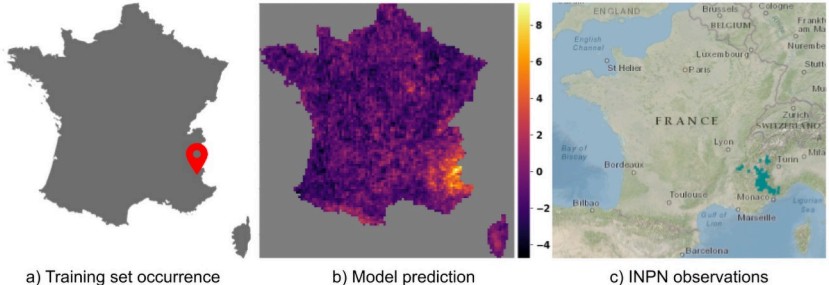

a) Training set occurrence
b) Model prediction
c) INPN observations

**Fig 7. Prediction of *Delphinium dubium* (Rouy & Foucaud) Pawl., 1934.** Compare the model prediction to the French National Inventory of Natural Heritage (INPN) occurrences map for *Delphinium dubium* (Rouy & Foucaud) Pawl., 1934.

one occurrence was used for training the response functions of that species). For *Senecio cacaliaster* the CNN-SDM model seems to have somewhat over-generalized the realized niche. Indeed, it can be seen a little activation in the Pyrenees mountains whereas the INPN occurrences, in accordance with the knowledge on this species, only fall in the central massif. This over-generalization, however, illustrates how the CNN-SDM model is capable of transferring knowledge across species, even with an extremely low number of occurrences.

## High variability of AUC and TSS

As explained earlier, in addition to the primary evaluation metric, we evaluated two other more traditional metrics, the area under curve (AUC) and the true skill statistics (TSS). This additional assessment was performed only for the CNN and the RF model since the performance of the two other models were significantly lower with the primary metric.

Table 2 and Fig 8 summarizes the results of AUC and TSS evaluations for the CNN-SDM and RF. We can see in Table 2 that the MS_AUC is very similar for both models with a score around 0.80. However, Fig 8 shows that there is a high variability across the species. The majority of species have very high AUC value but many species also have low AUC values, sometimes below 0.5. It is important to note here that 36% of the species in the test set only have 1 occurrence. For such species the AUC can only be 0.0 or 1.0 and thus has a very high variance. Similarly, species with 2 or 3 occurrences in the test are numerous (30%) and have a high variability in the estimation of the AUC. If we compare the boxplots of the AUC scores for the CNN and RF, we can see that the median, first quartile and last quartile are better for the CNN. Only the outliers lead to lower AUC values for the CNN and tend to bring the average value to that of RF. Similar conclusions can be formulated for the TSS as shown in Fig 8. For most species, the TSS score can only take a few possible values (1.0, 0.5, 0.0 or -1.0) which explains why the boxplots of both the CNN model and the RF model are aligned with these values.

**Table 2. CNN-SDM and RF evaluations with AUC and TSS.**

| Metric | CNN-SDM | RF |
|---|---|---|
| MS_AUC | 0.818 | 0.808 |
| MS_TSS | 0.450 | 0.459 |

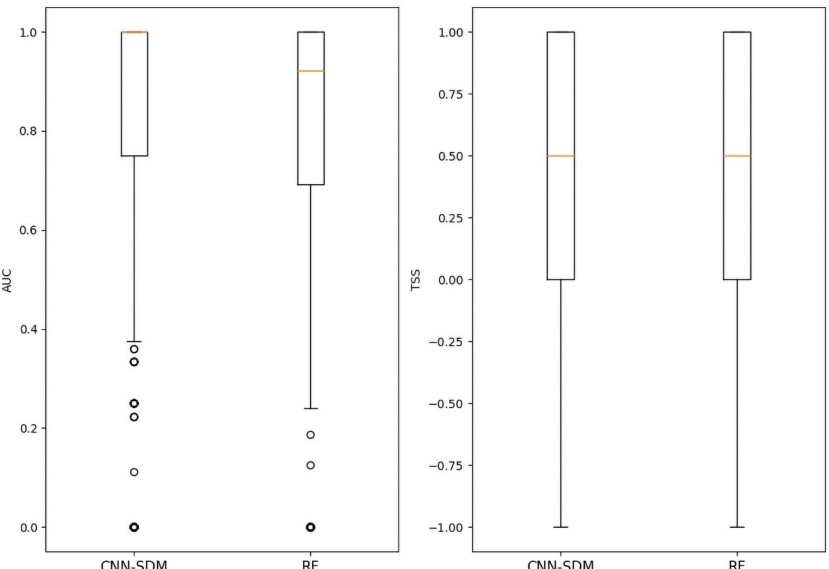

**Fig 8. AUC per species.** Boxplots of AUC and TSS of CNN-SDM, and RF for all species.

## Acknowledging spatial structure of environmental tensors in CNN-SDM improves performance

Fig 9 shows the results of ablation tests. The same CNN architecture was used to train models on different tensors transformations keeping or not the environmental variation and its environmental structure in the neighborhod of each location. The best model is the one learnt on the original tensors keeping these information. In addition, the model learnt on rotated tensors performed a little less well up to the $MSA_{40}$, but was similar above. Behind these two models,

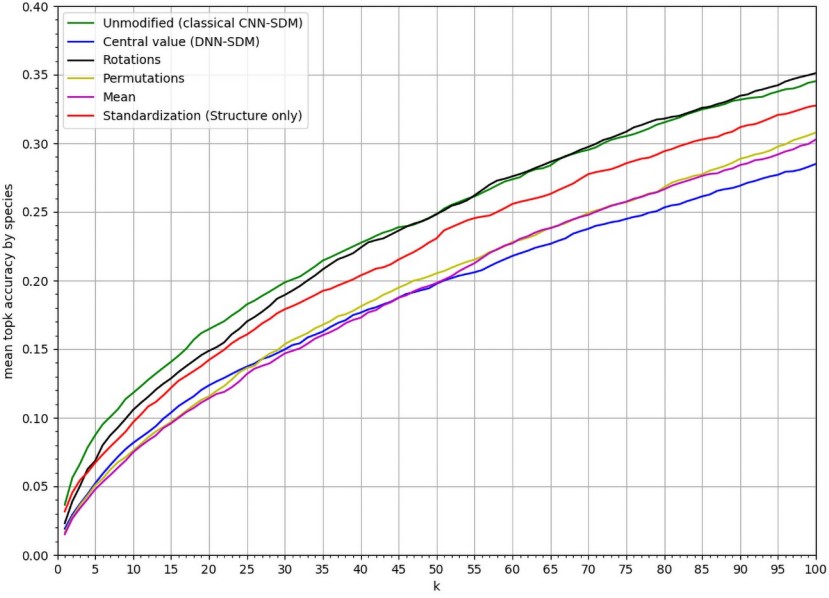

**Fig 9. Results of ablation study.** Results of the CNN with different tensor transformations in the ablation study: $MSA_k$ of the ablated CNN-SDM model deprived for $k$ values within [1, 100].

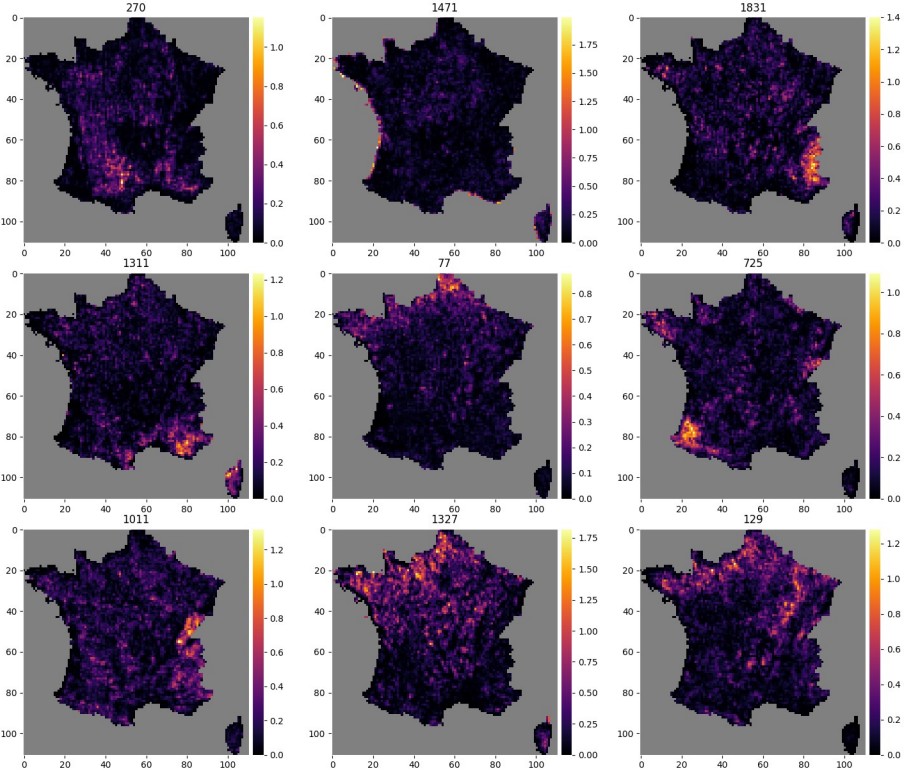

**Fig 10. Activations maps highlighting large scale geographical shapes.** Example of 9 activation maps, for 9 different neurons, among the 2048 neurons of the final layer of the CNN-SDM that shows large scale geographical shapes. The number of the neuron is indicated above each map.

the model based on standardized tensors was less performing but still better than the three last ones. The three worst performing models were those based on constant tensors, on permuted tensors and the DNN.

## Activation maps

Figs 10 and 11 each show the activation map of 9 neurons. Fig 10 shows activation maps of neurons with diverse spatial signatures. The activation of these neurons is spatially circumscribed to specific geographical areas, reflecting macroenvironmental or biogeographical Features. For examples neuron 1831 is activated in the Alps, neuron 1471 on coastal areas, and neuron 77 in the northernmost region. Fig 11 in contrast shows activation maps of neurons more difficult to interpret.

## Discussion

This study focuses mainly on the analysis of Convolutional Neural Network (CNN) models for the prediction of plant distribution. To be sure that this analysis is of interest we first validate our model against other more used models such as Boosted Trees (BT) and Random Forest (RF). We also compare the CNN to a similar Deep non-convolutional Neural Network (DNN). The main metric proposed is the mean *top-k accuracy* per species. This metric is adapted to our dataset that is presence-only, large scale (large area and numerous species) and highly unbalanced between species according to a long tail distribution. We also provide the

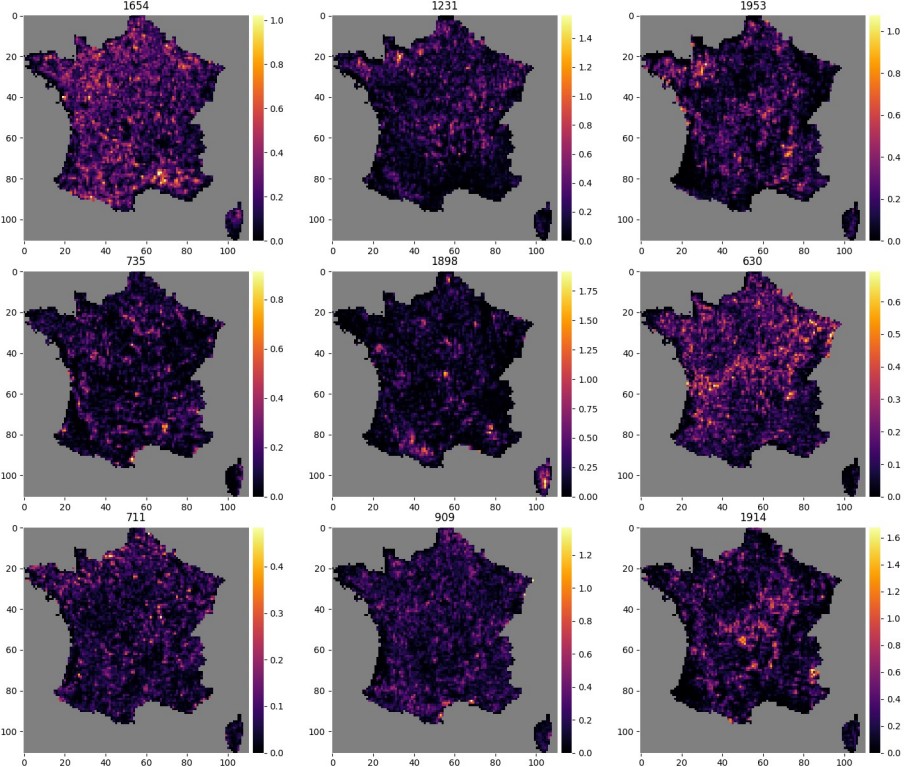

**Fig 11. Activations maps that do not highlight a particular geographical shape.** Example of 9 activation maps, for 9 different neurons, among the 2048 neurons of the final layer of the CNN-SDM that do not highlight a particular geographical shape. The number of the neuron is indicated above each map.

evaluation of the CNN and the best ponctual model, the RF, on more usual metrics, AUC and TSS. The main limitations of these metrics are (i) the bias induced by the generation of the pseudo-absence data and (ii) the irrelevance of the assessment for the species with the fewest occurrences, particularly those with only one or two occurrences. The results of CNN and RF were close using these metrics. In particular, CNN and RF achieved similar scores for the mean AUC over all species. The boxplot of the species-wise AUC values revealed that CNN is indeed better for most species but that outliers tend to degrade the mean value. These results highlight that the usual metrics for SDM (AUC, TSS, etc.) are not necessarily the most suitable for the type of model we are studying. The choice of pseudo-absences can artificially affect the score of models. In our context this choice is particularly difficult, the sampling effort is highly non uniform spatially and the long tail distribution results in very few observations for lots of species. Our choice of pseudo-absences is made in order to have less bias as possible but the remaining bias are still difficult to evaluate. In particular, the choice of other species occurrences as pseudo-absences can affect more the species with large distributions or the species present in habitat with high species richness. In the context of multi-species models these evaluations may penalize a model whose strength is to identify coherent groups of species. In addition, our models are designed for presence-only data and optimized to predict categorical conditional probabilities and not presence probabilities. This can be easily adapted for the CNN due to its functioning which produces an individual linear model for each species at the last layer of the network. This layer of logits then allows a prediction for each species that depends on the other species globally (through multi-species learning) but does not depend

directly on the prediction of the other species at a given spatial point. This is not possible for the RF model which is a classifier and which directly returns the relative probabilities. To limit this effect it is possible to use scaling to return predicted values between 0 and 1 for each species but the values obtained remain dependent on the values of the other species. For all these reasons we then choose to use the mean *top-k accuracy* per species as the main evaluation metric for the type of models studied. This metric has two main advantages: (i) it is not biased by the spatial distribution of the observation effort (because it is based on the species probability conditionally to an observation) and (ii), it allows evaluating the ability of the model to predict coherent groups of species jointly. We believe this metric is also not perfect either because the size of the set of species observable at a given location can be variable. Therefore, in future work, we plan to work on a more adaptive version of sets prediction evaluation (as for instance studied in [43]).

Our experiments first confirmed previous results of the literature that CNN-SDMs perform better than state-of-the-art methods such as boosted trees or random forest, but also than Deep Neural Networks using punctual environmental information (DNN-SDMs). DNN-SDMs were based on the same architecture than CNN-SDMs, but were blind to environmental neighborhood in the landscape surrounding occurrence points. Therefore, the environmental neighborhood more than the punctual environment matters for prediction. The performance is especially greater for rarer species in the dataset, while CNN-SDMs were less efficient than boosted tree and random forest for more frequent species. This central result is of both theoretical and practical interest, as rare species (often with narrow geographical distributions and specialized habitat requirements [44]) are more numerous, notoriously more difficult to predict, and important for conservation and management. There is a long debate on the way spatial autocorrelation in species distributions arises from both environmental structure and species dynamics [45, 46], and on how space should be acknowledged in the analysis of biodiversity patterns [47]. The CNN-SDM is based on environmental tensors, which represent not only the punctual environmental conditions of the sampled sites, but also the surrounding environmental conditions and their spatial structuring. To better understand what information contained in the surrounding environment allows a performance gain to the CNN-SDM we designed an original benchmark of alternative CNN-SDMs based on transformed tensors, each discarding either landscape-level spatial structure or heterogeneity in environmental factors. The comparison of these CNN-SDMs allowed identifying which information contained in the tensors improved predictive power. CNN-SDM calculated on unmodified tensors outperformed all CNN-SDMs learnt on transformed tensors. In particular, the CNN-SDM calculated on unmodified tensors is better than the CNN-SDMs learnt on permuted or averaged tensors. The comparison underlines that not only the average or the variance of the environment in the landscape, but also its spatial structuring matters. This result is confirmed by the model learned on standardized tensors (structure) which is better than the models learned on permuted or averaged tensors showing that the spatial structure is equally, if not more important than the value of the environment for the model. The fit of CNN-SDMs on real tensors could thus grasp the landscape-level influence of both environmental local values and spatial structuring. CNN-SDMs could get significant predictive power by acknowledging spatial structuring of environment around locations, *i.e.*, the local landscape structure.

These results support the role of landscape-level ecological processes in shaping species distributions. Specifically, the spatial structure of habitat fragmentation [48] and the amount of favorable habitat [49] in the landscape can both influence population persistence at given sites [50]. Better predictive ability of CNN-SDMs indeed supports the role of such landscape-scale drivers on species occurrences. For instance, for a binary predictor, the average in the landscape could represent available habitat amount, while the spatial structure of the landscape

could acknowledge the role of connectivity. Using different landscape configurations in CNN could help testing these alternative hypotheses and underline the potential of the approach for testing theories in spatial ecology.

Only the performance of the model on rotated images is close to the one of the CNN-SDM based on real environmental tensors. We can however note a difference between $MSA_1$ and $MSA_{40}$ where the model with rotated image is worse. Our interpretation is that the landscape-level orientation (*e.g.* north slope *vs.* south slope) has an impact on some species but is not the most important structural information contained in the tensors.

The classification schemes built by machine learning approaches incorporate recurrent motifs of species occurrences and the shared influences of primary environmental variable. It thus allows addressing the signatures of ecological processes on species assemblages, in line with alternative approaches such as jSDM [27]. SDM approaches classifying multiple species can incorporate macroecological constraints and acknowledge saturation rules [51, 52]. In addition, multi-species SDMs should be more robust to biases in occurrence information [24]. We notice that the problem addressed here, *i.e.* predicting species relative probabilities given that there is an occurrence, is different to predicting the relative occurrence intensity of each species across space, like it is done by MAXENT [9] and more generally Poisson Point Processes models [53]. The occurrence intensity estimated in the latter case is sensitive to the observation effort bias [25], while in the former case, we don't aim at providing a spatial intensity estimation. DNN and CNN are powerful approaches able to grasp complex influences of environmental variables on many species. Despite this complexity, regularisation rules have proved successful in selecting relevant information and parsimonious enough models. Therefore, CNN-SDMs should be able to grasp meaningful ecological and biogeographical patterns shared by many species, and thereby provide robust predictions.

The activation maps of the neurons of the last layer, *i.e.* the features, allow the visualization of the ecological patterns learned by the CNN-SDM. We found that neurons were active in relatively large or multiple areas, which could represent complex environmental and macroecological signatures corresponding to local or landscape-level environmental conditions. By nature, these integrative neurons could combine multiple environmental drivers and thus their complex and joint influence, *e.g.* through compensation processes. Some neuron activation maps were consistent with large-scale geomorphological patterns such as mountain ranges or coastal zones. The activation maps could be used to examine how emergent macroecological patterning stems from species dynamics and environmental variation [54]. It shows the potential of machine learning approaches to unravel large-scale macroecological patterns from intensive occurrence datasets [55].

Our study shows the benefit of using Convolutional Neural Networks for species distribution modelling (CNN-SDMs). First, their architecture allows learning highly non-linear environmental descriptors. Second, they are particularly effective for predicting distributions of rare species. Third, a major advantage is the ability of CNN to use very high dimensional data such as environmental tensors. Indeed, our study shows that the CNN-SDMs capture an information of environmental landscape structuring through environmental tensors. This information is richer than the punctual environment but is not accessible to conventional models.

## Supporting information

**S1 Protocole. Dataset protocole.** Detailed dataset construction protocole.
(PDF)

## Author Contributions

**Conceptualization:** Benjamin Deneu, Maximilien Servajean, Pierre Bonnet, Christophe Botella, François Munoz, Alexis Joly.

**Data curation:** Benjamin Deneu, Maximilien Servajean, Christophe Botella.

**Funding acquisition:** Pierre Bonnet, Alexis Joly.

**Investigation:** Benjamin Deneu, Maximilien Servajean, Pierre Bonnet, Christophe Botella, François Munoz, Alexis Joly.

**Methodology:** Benjamin Deneu, Maximilien Servajean, Christophe Botella, Alexis Joly.

**Project administration:** Pierre Bonnet, François Munoz, Alexis Joly.

**Software:** Benjamin Deneu, Maximilien Servajean.

**Supervision:** Maximilien Servajean, Pierre Bonnet, François Munoz, Alexis Joly.

**Validation:** Maximilien Servajean, François Munoz, Alexis Joly.

**Visualization:** Benjamin Deneu, François Munoz, Alexis Joly.

**Writing – original draft:** Benjamin Deneu, Maximilien Servajean, Pierre Bonnet, François Munoz, Alexis Joly.

**Writing – review & editing:** Benjamin Deneu, Maximilien Servajean, Pierre Bonnet, Christophe Botella, François Munoz, Alexis Joly.

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
