## [Decision Letter · Decision Letter 0]

2 Oct 2020

Dear Mr Deneu,

Thank you very much for submitting your manuscript "Convolutional neural networks improve species distribution modelling by capturing the spatial structure of the environment" for consideration at PLOS Computational Biology.

As with all papers reviewed by the journal, your manuscript was reviewed by members of the editorial board and by several independent reviewers. In light of the reviews (below this email), we would like to invite the resubmission of a significantly-revised version that takes into account the reviewers' comments.

We cannot make any decision about publication until we have seen the revised manuscript and your response to the reviewers' comments. Your revised manuscript is also likely to be sent to reviewers for further evaluation.

Sincerely,

Arne Elofsson

Deputy Editor

PLOS Computational Biology

Arne Elofsson

Deputy Editor

PLOS Computational Biology

Reviewer's Responses to Questions

**Comments to the Authors:**

Reviewer #1: This paper describes a deep learning approach to species distribution prediction that uses spatial environmental information as input to a convolutional neural network (CNN). The study is well carried out with ablation studies and comparison to other methods. The two main problems with the paper are 1) the exposition could be more clear (see below) and 2) other works [19,21] have already applied CNNs to this problem. To address 1) it is suggested that the authors work more on the writing (perhaps consult a native English speaker). On 2) it should be made more clear what the novelty is of this work.

Minor comments:

1. the later -> the latter

2. Define punctual model properly first time used.

3. For practical reasons 6 (data required), most Species Distribution Models (SDMs) are correlative methods 7 relating known species occurrence data to potential environmental predictors [2–7]. 8 Popular examples of such methods include MAXENT [8–10], random forest [11] and 9 boosted regression trees [12–14].

MaxEnt etc are general regression method that can be applied to SDM.

4. Eq (1) needs more explanation.

5. The paragraph Predictions is unclear. What is exactly being predicted? I understand a softmax is being used. So it is multinomial classification. But the data is presumably occurrence of each species. What is the conversion here?

6. There are also other examples similar to 4. and 5. later in the paper in the same spirit that it should be possible for the authors to spot without be pointed to by a referee. ;-)

Reviewer #2: In this paper, the authors compared convolutional neural networks (CNNs), deep but non-convolutional neural network models (DNNs), boosted trees (BT) and Random Forest (RF) for predicting species distributions. They found that CNNs outperformed the other models for rare species. This demonstrated the usefulness of CNNs. The authors used top-k accuracy to characterize the performance of the models. However, this approach of model evaluation is difficult for species distribution modelling practitioners to understand because they always use the conventional model accuracy measures (including the area under the receiver’s operating characteristic curve, true skill statistic, sensitivity and specificity, etc.). It’s better also to give the model evaluation using these measures.

Spefics:

P2 L71: Table ??

P7 L193: “is apply at” may be changed to “is applied at”?

P8 L214: “are uses” may be changed to “are used”?

P10 L298-299: “the environmental neighborhood more than the punctual environment matters for prediction”?

P10 L309: “it’s”?

P11 L356: change “parcimonious” to “parsimonious”

Some information is lost for this reference:

52. Botella C, Joly A, Monestiez P, Munoz F, Bonnet P. Bias in presence-only niche models related to sampling effort and species niches: lessons for background point selection. 2020;.

**Have all data underlying the figures and results presented in the manuscript been provided?**

Reviewer #1: Yes

Reviewer #2: Yes

PLOS authors have the option to publish the peer review history of their article (what does this mean?). If published, this will include your full peer review and any attached files.

Reviewer #1: No

Reviewer #2: No
---

## [Decision Letter · Decision Letter 1]

8 Mar 2021

Dear Mr Deneu,

We are pleased to inform you that your manuscript 'Convolutional neural networks improve species distribution modelling by capturing the spatial structure of the environment' has been provisionally accepted for publication in PLOS Computational Biology.

Best regards,

Arne Elofsson

Deputy Editor

PLOS Computational Biology

Arne Elofsson

Deputy Editor

PLOS Computational Biology

Please make sure that the minor changes suggested by Reviewr #2 are corrected in the proofs

Reviewer's Responses to Questions

**Comments to the Authors:**

Reviewer #1: The authors have substantially improved the paper and addressed the concerns of the reviewers in good manner as far I can see. I leave it to the other reviewer to address whether the evaluation metrics question has been satisfactory answered.

Detailed comments:

1. The figure should be made higher resolution.

2. Add zoomed in plots.

Reviewer #2: Comments to the authors

I am satisfied with the authors’ response. Now, I only have a minor concern. In P11 L352-353: The authors said: “It is important to note here that 36% of the species in the test set only have 1 occurrence. For such species the AUC can only be 0:0 or 1:0 and thus has a very high variance.” This is not correct. Even though there is only one occurrence, you have many (pseudo-)absences. Assume the prediction for the occurrence is p1 (between 0 and 1), if the predictions for all the (pseudo-)absences are less than p1, AUC = 1; if the predictions for all the (pseudo-)absences are larger than p1, AUC = 0; if the predictions are less than p1 for some (pseudo-)absences and are larger than p1 for the other (pseudo-)absences, 0 < AUC < 1. For example, if p1 = 0.8 (for the occurrence) and p0 = c(0, 0.1, 0.2, …, 0.9, 1) (for 11 (pseudo-)absences), AUC = 0.2273. The AUCs may have a very high variance, but they can be any value between 0 and 1. This is also true for TSS.

Specifics:

P1: (in Abstract) “Convolutional Neural Networks (CNNs) are statistical models suited for learning

complex visual patterns.” They are machine learning methods.

P3 L71: In “We use 33 environmental raster variables”, may use “used”?

**Have all data underlying the figures and results presented in the manuscript been provided?**

Reviewer #1: Yes

Reviewer #2: Yes

PLOS authors have the option to publish the peer review history of their article (what does this mean?). If published, this will include your full peer review and any attached files.

Reviewer #1: **Yes: **Ole Winther

Reviewer #2: No

---

## [Editor Report · Acceptance letter]

16 Apr 2021

PCOMPBIOL-D-20-00821R1 

Convolutional neural networks improve species distribution modelling by capturing the spatial structure of the environment

Dear Dr Deneu,

I am pleased to inform you that your manuscript has been formally accepted for publication in PLOS Computational Biology. Your manuscript is now with our production department and you will be notified of the publication date in due course.

With kind regards,

Katalin Szabo
